# Prediction of Rapid Early Progression and Survival Risk with Pre-Radiation MRI in WHO Grade 4 Glioma Patients

**DOI:** 10.3390/cancers15184636

**Published:** 2023-09-19

**Authors:** Walia Farzana, Mustafa M. Basree, Norou Diawara, Zeina A. Shboul, Sagel Dubey, Marie M. Lockhart, Mohamed Hamza, Joshua D. Palmer, Khan M. Iftekharuddin

**Affiliations:** 1Vision Lab, Department of Electrical & Computer Engineering, Old Dominion University, Norfolk, VA 23529, USA; wfarz001@odu.edu (W.F.); shboulzeina@gmail.com (Z.A.S.); 2Department of Internal Medicine, OhioHealth Riverside Methodist Hospital, Columbus, OH 43214, USA; mustafa.basree@ohiohealth.com (M.M.B.); sagel.dubey2@ohiohealth.com (S.D.); 3Department of Mathematics & Statistics, Old Dominion University, Norfolk, VA 23529, USA; ndiawara@odu.edu; 4OhioHealth Research Institute, Columbus, OH 43214, USA; marie.lockhart@ohiohealth.com; 5Department of Neurology, OhioHealth, Columbus, OH 43214, USA; mohamed.hamza@ohiohealth.com; 6Department of Radiation Oncology, The James Cancer Hospital and Solove Research Institute, Ohio State University Wexner Medical Center, Columbus, OH 43210, USA; joshua.palmer@osumc.edu

**Keywords:** rapid early progression (REP), pre-radiation MRI, radiomics, glioblastoma (GB), machine learning (ML), survival analysis, dependent censoring, copula modeling

## Abstract

**Simple Summary:**

Rapid early progression (REP) has been defined as increased nodular enhancement at the border of the resection cavity, the appearance of new lesions outside the resection cavity, or increased enhancement of the residual disease after surgery and before radiation. Patients with REP have worse survival compared to patients without REP (non-REP). Therefore, a reliable method for differentiating REP from non-REP is hypothesized to assist in personlized treatment planning. A potential approach is to use the radiomics and fractal texture features extracted from brain tumors to characterize morphological and physiological properties. We propose a random sampling-based ensemble classification model. The proposed iterative random sampling of patient data followed by feature selection and classification with radiomics, multi-resolution fractal, and proteomics features predicts REP from non-REP using radiation-planning magnetic resonance imaging (MRI). Our results further show the efficacy of pre-radiation image features in the analysis of survival probability and prognostic grouping of patients.

**Abstract:**

Recent clinical research describes a subset of glioblastoma patients that exhibit REP prior to the start of radiation therapy. Current literature has thus far described this population using clinicopathologic features. To our knowledge, this study is the first to investigate the potential of conventional radiomics, sophisticated multi-resolution fractal texture features, and different molecular features (MGMT, IDH mutations) as a diagnostic and prognostic tool for prediction of REP from non-REP cases using computational and statistical modeling methods. The radiation-planning T1 post-contrast (T1C) MRI sequences of 70 patients are analyzed. An ensemble method with 5-fold cross-validation over 1000 iterations offers an AUC of 0.793 ± 0.082 for REP versus non-REP classification. In addition, copula-based modeling under dependent censoring (where a subset of the patients may not be followed up with until death) identifies significant features (*p*-value < 0.05) for survival probability and prognostic grouping of patient cases. The prediction of survival for the patients’ cohort produces a precision of 0.881 ± 0.056. The prognostic index (PI) calculated using the fused features shows that 84.62% of REP cases fall under the bad prognostic group, suggesting the potential of fused features for predicting a higher percentage of REP cases. The experimental results further show that multi-resolution fractal texture features perform better than conventional radiomics features for prediction of REP and survival outcomes.

## 1. Introduction

Brain and other central nervous system (CNS) tumors are associated with the highest mortality and morbidity across different malignancies in the United States [1]. Glioblastoma (GB) is one of the most aggressive brain tumors, representing nearly half of brain gliomas [2]. For newly diagnosed GB patients, the standard of care includes maximal safe surgical resection followed by radiation therapy with concurrent and adjuvant temozolomide (TMZ) after which tumor-treating fields are often recommended (i.e., the Novo-TTF system, renamed Optune) [3]. Radiation therapy should ideally begin six weeks following surgery [4]. During this time frame, GB may regrow significantly due to its highly proliferative nature [5]. Several institutional series have been published evaluating post-operative REP with pre-radiation MRI [6,7,8]. Specifically, REP is assessed by comparing early post-operative MRI scans with radiation-planning MRI [8]. There is almost a 50% prevalence rate for the development of REP, even if radiation is initiated earlier than six weeks post-operatively [6,7,8].

MRI plays a crucial role in the evaluation of post-operative and post-treatment effects. The post-operative MRI is acquired within 3 days (preferably within 24 h) both to assess extent of resection and to minimize the effect of enhancement due to surgery [5]. The radiation-planning MRI scan is obtained 1 to 3 weeks prior to the start of radiation therapy to assist with target delineation. According to the guidelines [9], another MRI (post-radiation) scan is obtained 2 to 6 weeks after the completion of treatment with radiation therapy +/− TMZ followed by surveillance imaging every 2–4 months. The comparison of the first post-radiation MRI (approximately after 1 month from the completion of radiation therapy) to baseline scan (post-operative MRI, more commonly being radiation-planning scan), is often made to evaluate tumor progression as well as radiation-induced changes (collectively termed as pseudo progression) [4]. MRI offers a detailed characterization of the morphological, physiological, and metabolic properties of brain tumors, particularly brain gliomas, which are complex and heterogenous malignancies [10]. Quantitative radiomics features [11,12] extracted from MRI (texture, intensity, shape, area, and geometric features) followed by statistical and machine learning analyses have been shown to be effective [13,14,15] in brain tumor volume segmentation, and classification of normal/tumor tissues.

Patients with REP have worse survival compared to non-REP patients [7]. Overall survival (OS) analysis refers to the time to death from the day of surgery. A patient is censored if they are lost to follow-up prior to observing time to death [16]. Censoring may introduce bias into statistical analysis results if censoring processes involve dropout or withdrawal due to tumor progression, treatment toxicity, or the start of second-line therapy [16]. Because a patient may die soon after being lost to follow-up, total survival and dropout time may be positively associated [16]. Dependent censoring occurs when the relationship between censoring time and survival time cannot be explained by observable factors [16,17]. In statistical analysis, copula-based modeling is a state-of-the-art method to model the dependency between survival and censoring time. 

As alluded to above, several retrospective reviews correlated clinical and pathologic features with REP, and found it to be an independent negative prognostic factor [4,6,7,8]. It remains unclear whether patients with REP have distinct molecular or radiographic features. To the best of our knowledge, there has been no research that utilizes MRI features for prediction of REP using radiation-planning MRI scans, which may act as a quantitative imaging-based biomarker to stratify REP patients from non-REP patients.

The first objective of this study is to evaluate the predictive efficacy of conventional radiomics and sophisticated multi-resolution fractal texture features extracted from radiation-planning MRI for predicting REP. A second objective is to examine the survival probability of patients using copula modeling applied to radiation-planning MRI radiomics features within a context of dependent censoring scenario. Third objective is the binary prediction of patients’ survival status (patients expired/dead or not) based on selected significant (*p*-value < 0.005) T1C features, employing copula modeling. To assess the significance of the selected features, a prognostic index (PI) is calculated, derived as a linear combination of the selected features. By utilizing this calculated PI, patients are categorized into prognostic groups (good or bad). Additionally, within these prognostic groups, the distribution of REP cases versus non-REP cases is constructed. The distribution of REP cases versus non-REP cases within the defined prognostic groups further demonstrates the predictive capability of radiomics features in identifying REP as a bad prognostic factor or indicator of a high-risk group.

## 2. Materials and Methods

### 2.1. Patient Data

This is an institutional review board (IRB, reference #22-057)-approved retrospective chart review of a cohort of patients treated at OhioHealth between 1 January 2015 and 1 March 2021. Relevant clinical and radiographic data have been abstracted from the electronic health record system. Patients with a biopsy-confirmed diagnosis of World Health Organization (WHO) grade 3 or 4 anaplastic astrocytoma or grade 4 glioblastoma have been included in this study, with a minimum of three MRI scans (pre-operative, early post-operative, radiation planning). Imaging studies are reviewed by board-certified neuroradiologists. All patients have undergone surgery (biopsy, subtotal, or gross total resection), followed by radiation therapy with or without adjuvant TMZ.

A total of 95 patient cases have been included in this study. Among these, twenty-five cases do not have a (T1C) sequence in at least one of the MRI studies. Seventy patients with complete radiographic data have been included in the analysis, with thirteen of them being clinically identified as having REP and the remaining fifty-seven as non-REP. REP has been classified as such in line with previous literature [18,19]. A detailed summary of patient cases is presented in Table 1.

### 2.2. Algorithm Pipeline for Prediction of Rapid Early Progression (REP)

The overall pipeline for prediction of rapid early progression (REP) is depicted in Figure 1. The specifics of each step are described in the subsequent sub-sections. The predictive model is an ensemble tree-based Cat Boost classification model implemented in Python 3.7. The model’s performance is evaluated through 1000 iterations and 5-fold cross-validation, resulting in 5000 different values for computing the area under curve (AUC), positive predictive value (PPV), false positive rate (FPR), and accuracy. The statistical distribution of the 5000 values is considered for describing the model’s performance.

### 2.3. MRI Preprocessing, Tumor Volume Segmentation and Feature Extraction

#### 2.3.1. MRI Preprocessing

All radiation-planning MRI images are co-registered to the same T1 anatomic template using affine registration and resampled to 1 mm^3^ voxel resolution using the Oxford Center for Functional MRI of the Brain’s (FM-RIB’s) Linear Image Registration Tool (FLIRT) of the FMRIB Software(version 6.0) Library (FSL) [20]. The FSL’s Brain Extraction Tool (BET) [21] is utilized to skull-strip each patient’s volumetric image. For obtaining improved skull-stripped volumetric images, manual intervention is employed. Additionally, all images are smoothed using the Smallest Unvalued Segment Assimilating Nucleus (SUSAN) [22], a low-level image processing technique, in order to reduce high frequency intensity changes (i.e., noise) in regions with a uniform intensity profile while maintaining the underlying structure. The intensity histograms of all modalities for all patients are then matched to the relevant modality of a single reference patient using the implemented version of the Insight Toolkit (ITK) [23].

#### 2.3.2. Tumor Volume Segmentation

The utilization of transfer learning in our study is motivated by the inherently challenging nature of limited patient data for tumor volume segmentation. Given the scarcity of available data, we employ transfer learning as a strategic approach to enhance the performance of our segmentation model. To achieve this, we leverage a substantial dataset comprising 1251 GB patient cases collected from the well-established BRATS 2021 challenge [24,25,26]. This dataset serves as the foundation for training our segmentation model. By initially training the model on this larger dataset, we enable it to learn intricate features and patterns associated with tumor tissue segmentation. Once the model is adequately trained using this extensive dataset, it is subsequently deployed to segment tumor tissue regions in a more restricted 70 patient cohort with radiation-planning MRI volumetric images. This process allows the model to generalize its learned knowledge and effectively adapt to the specific characteristics of our target cohort, even in the presence of limited data. For the task of segmenting tumor tissue regions, encompassing edema, enhancing tumor, and necrosis, we employ a 3D UNet model specifically designed for GB patients’ T1C MRI scans [13,14].This approach capitalizes on the model’s inherent ability to capture complex spatial relationships within the volumetric images, further enhancing the precision and accuracy of tumor tissue segmentation. The tumor regions are rigorously validated by two expert radiation oncologists specializing in primary and secondary brain tumor treatment, who achieve consensus.

#### 2.3.3. Feature Extraction

A total of 600 features have been extracted from the tumor tissue volume segments in this study. These features include texture, volume, and the area of the tumor and its sub-regions (edema, enhancing tumor, and necrosis). Forty-one texture characteristics have been derived from the whole tumor volume in the raw MRI (T1C) sequence, as well as the tumor sub-regions. The conventional texture features are extracted using a grey-tone spatial dependence matrix (GTSDM), neighborhood grey-tone difference matrix (NGTDM), and grey level size zone matrix (GLZSM). The fractal texture features encompass the piecewise triangular prism surface area (PTPSA) for fractal characterization, multi-resolution Brownian motion (mBm) analysis, and tumor region characterization with Holder Exponent (HE) modeling, termed as generalized multi-resolution Brownian motion (GmBm). The PTPSA, mBm, and HE computational algorithms are detailed in [27,28,29]. Multi-resolution fractal features depict textural variation in tumor tissue across various image resolutions [30]. Six histogram-based statistics (mean, variance, skewness, kurtosis, energy, and entropy) are also derived from the distinct tumor sub-regions. We further extract volumetric features: the volume of the entire tumor, the volume of the whole tumor in relation to the brain, and the volume of sub-regions. Vallières et al.’s [31] MATLAB-based software (version R2019b) is utilized for analyzing texture features. For fractal characterization, multi-resolution fractal characterization, HE characterization, and volumetric characteristics, MATLAB-based in-house software is employed.

### 2.4. Selection of Radiomics Features and Model Building

To evaluate the efficacy of fractal features and conventional radiomics features, we consider two model configurations. These model configurations include (a) a non-fractal model (a model containing only conventional volume, area, and texture features), and (b) a fractal model (a model incorporating multi-resolution fractal features along with conventional volume, area, and texture features) [32]. For feature selection, a two-step feature selection process (details of feature selection and statistical analysis can be found in Appendix A) results in three significant (*p*-value < 0.05) features for both the fractal and non-fractal models, respectively. Radiomic modeling of REP classification using the selected features is implemented using a nested paradigm involving 1000 iterations and 5-fold cross-validation with random sampling (details of the algorithms are provided in Appendix A). The objective of cross-validation is to offer a robust estimate of a model’s performance on unseen data [33]. In a k-fold cross-validation, the data are partitioned into k-subsets, of approximatively equal sizes. Moreover, typically calculated from the formula: n=k×m, where n is the sample size, k is the number of folds, and m the number of observations in each fold or subset. For 70 patient cases, where *n* = 70, *k* = 5, and *m* = 14, 70 = 5 × 14, we see that 5-fold may be a reasonable choice for the fold numbers.

### 2.5. Survival Analysis Modeling under Dependent Censoring

The Kaplan–Meier estimator and the Cox proportional hazard model are typical methods for survival analysis and feature selection [17]. These techniques manage censoring under the presumption that overall survival time and censoring time are statistically independent. Therefore, a copula-based approach [17] is employed to estimate the dependence parameter by utilizing cross-validation to select significant genes or features for survival prediction. The dependency between survival and censoring time is modeled by copula functions.

Considering the censoring scenario as presented in Table 2, we propose the copula method (details are presented in Appendix B) for feature selection in survival analysis. First, we identify the dependency parameter utilizing the survival data and the extracted radiomics features matrix. In addition to imaging features, molecular features are also included in the feature matrix. The significant (*p*-value < 0.05) features are then utilized to make binary predictions regarding whether the patient has expired or is alive.

### 2.6. Survival Prediction

Figure 2 illustrates the comprehensive pipeline for predicting overall survival (OS). The objective is to predict the status of patients who have expired. Given that the copula-based method is computationally demanding [33], we perform feature selection in two steps. In the first step, we use our proposed algorithm (shown in Algorithm A1 in Appendix A). Subsequently, based on the F1-score range (0.2–0.84), we initially select features with F1-scores greater than 0.7. The number of selected features in the first step is 124 and 87 for the non-fractal and fractal models, respectively. These initially selected features serve as input for a second-step feature selection using the copula model, with the aim of identifying significant features (*p*-value < 0.05). These final selected features are associated with patients’ OS probabilities. Additionally, we examine whether the inclusion of molecular features (MGMT status, IDH status) is statistically significant in relation to patient survival probability. 

## 3. Results

### 3.1. Predictive Performance of Rapid Early Progression (REP) Classification

The comparative performance of the two model configurations is illustrated in Table 3. When we compare the performance of the fractal model to the non-fractal model, the fractal model attains an AUC of 0.793, while the non-fractal model attains an AUC of 0.673. A significant difference (determined through ANOVA tests, *p*-value < 0.001) exists in terms of accuracy, AUC, PPV, and FPR between the two model configurations, as presented in Figure 3.

The statistically significant features (please refer to Table A1 in Appendix A) in the non-fractal model are the eccentricity in the edema region, the second axis (*y*-axis) length in the necrosis region and the autocorrelation of GTSDM from T1C, respectively. For our statistical analysis, a *p*-value ≤ 0.05 is considered significant. We observe the significant differences in the features between the two groups. For instance, the median and mean value of the necrosis region for the REP group are higher than those of the non-REP. A similar trend is also observed in the autocorrelation of GTSDM from T1C between the two groups (please refer to Table A3 and Figure A2 in Appendix A).

For the fractal model, the statistically significant features (please refer to Table A2 in Appendix A) are GmBm of the NGTDM of T1C, the strength of NGTDM from the 37th direction of T1C, and the strength of NGTDM. We observe that the feature distribution is not normal in each group. Therefore, the Wilcoxon–Mann–Whitney test is performed to determine the significant difference (*p*-value < 0.05) in the feature distributions between the two groups. In the fractal model, the significant selected features comprise texture features. The median value of the selected features is higher in the non-REP group compared to the REP group (please refer to Table A4 and Figure A3 in Appendix A). 

### 3.2. Survival Probability Analysis under Dependent Censoring

First, we analyze the impact of dependent censoring on feature selection for survival probability. For this purpose, we evaluate the significant (*p*-value < 0.05) features using the Cox proportional hazard model with independent censoring (please refer to Table A5 in Appendix B). The features selected with independent censoring are compared with those selected with dependent censoring. Table 4 presents the significant features (*p*-value < 0.05) obtained with dependent censoring using copula modeling. In the case of non-fractal models, no significant features are selected when applying Cox modeling. To examine the effect of dependent censoring on feature selection, we compare the survival probability curves utilizing the top two features in the fractal model. Since only two features are selected with independent censoring, we proceed to analyze the survival marginal curves with and without censoring dependency.

For a patient case with feature vector $x=(x_{1}$,$x_{2},\ldots .,x_{p}$)′, survival prediction is analyzed using the PI defined as $\hat{\beta {\left(\alpha \right)}^{\prime }}x$, where $\hat{\beta {\left(\alpha \right)}^{\prime }}$ $=(\hat{\beta_{1}}$ (α),…., $\hat{\beta_{p}}$ (α) [16,34]. If α = 0, then PI = $\hat{\beta {\left(0\right)}^{\prime }}x$ which is based on Cox modeling under independent censoring (α = 0). Therefore, the PI for the fractal modeling with the cox model with two significant features is the following.(1)PI (with independent censoring) = (5.46 × T1C_mBm_GLZSM_LargeZoneLowGrayEmphasis) + (2.46 × T1C_ptpsa_GLZSM_LargeZoneLowGrayEmphasis).

However, considering dependent censoring (α = 18) and utilizing the copula modeling the top two selected features as shown in Table 4 and the PI for the fractal model is the following:(2)PI (with dependent censoring) = (−1.58 × ET2) + (0.74 × L2_Orientation).

Using the PI, we randomly divide the 67 patient cases into two groups of equal sample size (n_1_ = 33, n_2_ = 34). Patients in the good(low-risk) prognostic group have low PIs, and patients in the bad (high-risk) prognostic group have high PIs [16,17,35]. The two survival curves are determined by the copula graphic (CG) estimator [36,37] with the Clayton copula as presented in Figure 4. The difference between the two curves is calculated by the average vertical difference [16,17,38]. The *p*-value is calculated between the two groups using 1000 permutation tests [16,17,39]. From Figure A5a, we observe that the vertical distance (D = 0.128) between the two groups with independent censoring (α = 0), is not significant (*p*-value = 0.1422). However, considering the dependent censoring in Figure A5b (α = 18, c-index= 0.519), the distance (D = 0.185) between two prognostic groups is significant (*p*-value = 0.0047). 

### 3.3. Binary Prediction of Survival

Through the analysis, we observe the effect of dependent censoring on feature selection and survival probability (please refer to Figure A4 in Appendix B). Therefore, for binary survival prediction, we utilize the features selected through copula modeling. Table 4 presents the significant (*p*-value < 0.05) features for the fractal and non-fractal models. For the experimental analysis of the selected features, we consider the top three, five, seven, and nine features for binary classification of survival (whether the patient is expired or not). In the case of the fractal model, 10 features are significant for survival probability analysis while for the non-fractal model, 8 features are significant as presented in Table 4. First, we analyze the vertical distance between the good prognostic and bad prognostic groups with 3, 5, 7, 9, and 10 features (please refer to Table A6 in Appendix B). Subsequently, based on the significance of the feature combinations, we compute binary predictions of survival.

Using the selected top three, five, seven, and nine features, we apply our proposed algorithm (refer to Algorithm A2 in Appendix A) for binary prediction of survival. The predictive results for 5-fold cross-validation with 1000 iterations are presented in Table 5. In the case of binary survival prediction, we emphasize the precision of the models for the selected number of features. Higher precision corresponds to a lower false positive rate, as indicated in Table 5. Moreover, following the algorithm (refer to Algorithm A2 in Appendix A), we consider balanced number of expired/dead (*n* = 25, randomly sampled from 54 dead cases) and not dead (*n* = 13) cases in each iteration. It is observed (please refer to Appendix B) that when utilizing seven features, both model configurations (fractal, non-fractal) achieve a higher PPV or precision. Therefore, the performance of the model configurations is based on the top seven features. While comparing the fractal and non-fractal model performance for binary survival prediction, a significant difference in the area under the curve (AUC) is observed with a *p*-value of 0.005 from ANOVA analysis. Furthermore, in terms of PPV (*p*-value < 0.01), accuracy (*p*-value < 0.001),and false positive rate (*p*-value < 0.01), a significant difference exists between the performance of the fractal and non-fractal models.

In addition to radiomics features, we analyze the survival probability and binary prediction of survival using molecular information (MGMT methylation status, IDH status). Therefore, with the top seven selected features we include MGMT status and IDH mutation status as additional features. In both the fractal and non-fractal models, the MGMT status does not hold significance (*p*-value = 0.9651) under dependent censoring copula modeling. However, the IDH mutation status is significant (*p*-value = 0.04). Consequently, we added the IDH mutation status to the top seven features and computed model performance, as presented in Table 6. With the inclusion of additional molecular features, the distance between marginal survival curves remains almost the same. The vertical distance is D = 0.171, with *p*-value = 0.0085, as depicted in Figure 5 for the fractal model. In the non-fractal model, the vertical distance is D = 0.173, with *p*-value of 0.0084.

For both the fractal-molecular and non-fractal-molecular models, there exists a significant difference between the models in terms of accuracy, PPV and FPR. However, there is no significant difference between the model configurations regarding AUC. The reason for this could be illustrated in Figure 5, where the survival probability demonstrates almost the same vertical distance and *p*-value for both models with the addition of the molecular features to the seven radiomics features. Moreover, with inclusion of molecular features, there is no increase in model performance in terms of PPV and FPR compared to model configurations without molecular feature.

### 3.4. Analysis of Prognostic Groups and Its Association with REP Status

Based on Table 4, it is observed that for the non-fractal model, a total of eight features are found to be significant (*p*-value < 0.05), whereas for the fractal model, the number is 10. To analyze the distribution of REP patients in prognostic (good or bad) groups, we consider the top eight features in both the fractal and non-fractal models. As a result, the PI for the fractal and non-fractal models are as follows:(3)PI (Fractal model)) = (−1.58 × ET2) + (1.33 × T1C_ptpsa_GLZSM_LGLZE) + (0.74 × L2_Orientation) + (−0.91*edema_First_Axis_Length) + (−0.93 × wt_Major_Axis_Length) + (0.76 × L1_Extent) + (0.70 × L3_Orientation) + (2.27 × T1C_mBm_GLZSM_LZLGE), and (4)PI (Non-fractal model) = (−1.58 × ET2) + (0.74 × L2_Orientation) + (−0.91 × edema_First_Axis_Length) + (−0.93 × wt_Major_Axis_Length) + (0.76 × L1_Extent) + (0.70*L3_Orientation) + (−1.06 × nec_Second_Axis_1).

The PI is computed using the selected radiomics and multi-resolution fractal features from radiation-planning MRI. The only difference between the PI of the fractal and non-fractal models lies in the inclusion of multi-resolution fractal texture features in the fractal models, in addition to conventional texture features. The sixty-seven patients are divided into good prognostic and bad prognostic groups. Using a CG estimator, two marginal survival curves are determined. In the fractal model, the distance between the two marginal curves is D = 0.157 (*p*-value = 0.014), while in the non-fractal model, it is D = 0.128 (*p*-value = 0.038).

Figure 6 displays the distribution of REP cases within the prognostic groups. The scatter points depict the prognostic index for each patient. The darker point indicates the mean PI within each group, while the bars represent the corresponding standard deviation. For instance, in Figure 6a, within the scatter plot, the group labeled “c” corresponds to the patients in the bad prognostic group, which also includes REP cases. As indicated by Figure 6, we can observe that patients with a low prognostic index (PI)/lower risk are placed in the good prognostic group while those with a higher PI/higher risk are placed in the bad prognostic group. Furthermore, in the case of fractal model, based on the PI, 84.62% (11 out of 13 cases) of REP cases fall under the bad prognostic group, as depicted in the matrix representation of Figure 6a. In the non-fractal model, 76.92% (10 out of 13 of REP cases belong to the bad prognostic group, as presented in in the matrix representation of Figure 6b.

The percentage of REP cases in the bad prognostic group is higher for the fractal model compared to the non-fractal model. Therefore, we have calculated whether there exists a significant difference between groups in terms of survival time, as illustrated in Table 7. In both groups, there is a significant (*p*-value < 0.05) difference in survival time. Test of normality and the appropriate test (ANOVA/Wilcoxon–Mann–Whitney) have been conducted between the two groups. From Table 7, the median number of survival days for REP is 172 days, in contrast to the non-REP group’s 474.50 days, highlighting the significant difference between their survival times. Moreover, within each prognostic group (good or bad), there is also a significant difference in survival days between the groups. The median survival days for the bad prognostic group is 329 days, which is significantly different from the median survival days of 511 days for the good prognostic group.

In addition, we have analyzed each individual significant feature of the fractal model in relation to the REP status. For each individual feature, a test of normality, specifically the Shapiro–Wilk test, has been performed, followed by an appropriate test (ANOVA/Wilcoxon–Mann–Whitney) to determine the significance of each feature with respect to the REP status. As depicted in Figure 7, it can be observed that two features from the fractal survival probability exhibit a significant association (*p*-value < 0.05) with the REP status. Among these two features, the fractal features belong to the top eight features of the fractal model. Therefore, the higher percentage of REP cases in the fractal model can be associated with the significance of this feature with the REP status. The multifractal feature from T1C is significantly linked to the dependent censoring survival probability and REP status.

## 4. Discussion

This study proposes the feasibilty of radiomics and sophisticated multi-resolutional fractal texture features for prediction of REP status in GB patients from a radiation-planning T1C sequence MRI. Two models (non-fractal and fractal) are constructed utilizing random sampling 5-fold crossvalidation as presented in Table 3. The predictive performance of the fractal model is an AUC of 0.793 ± 0.082, with an FPR of 0.145 ± 0.107, while that of the non-fractal model is an AUC of 0.673 ± 0.082 with an FPR of 0.262 ± 0.177. There is a significant difference (*p*-value < 0.001) between the fractal and the non-fractal models for the prediction of REP status.

Furthermore, copula-based modeling for survival analysis of dependent censoring has been obtained using survival time, censoring time, and radiomics features. For binary prediction of patient survival, the selected significant features (*p*-value < 0.05) from survival analysis have been incorporated. The predictive precision performance for patient survival in the fractal model is 0.881 ± 0.056, with an FPR of 0.311 ± 0.109, while that of the non-fractal model is 0.872 ± 0.054, with an FPR of 0.339 ± 0.106. Moreover, the inclusion of IDH mutation status in addition to radiomics features holds significance (*p*-value = 0.04) for the survival probability analysis of patients.

Note a direct comparison of our proposed method with the existing literature may not be feasible due to the differences in patient dataset and the analysis methods. The methods in the existing literature primarily focus on describing REP and assessing its impact on disease outcomes using pre-radiation MRI. Additionally, a limited number of studies [4,35] have concentrated on determining the significance of radiation-planning MRI for assessing pseudo-progression. Previous studies [7] investigated the intergration of REP and MGMT status for the overall survival of patients. Patients with both REP and unmethylated MGMT status exhibit a worse prognosis compared to non-REP and methylated patients (10.2 months versus 16.5 months, *p*-value = 0.033). 

In comparison to the existing literature on REP in pre-radiation MRI, our method focuses on evaluating the diagnostic and prognostic ability of radiomics features in both identifying REP in patients with glioblastoma, and predicting their survival outcomes under dependent censoring. We first extract radiomics features from radiation-planning T1C MRI and assess their ability to predict REP. Our analysis shows that the inclusion of multi-resolution fractal features improves model performace significantly (*p*-value < 0.05) when compared to the non-fractal feature model. We then carry out survival anlysis utlizing copula-based modeling of the occurrence of dependent censoring. The CG estimator is used to straify good and bad prognostic groups based on radiomics and multi-resolution fractal feature-based PI. Median survival in days is higher for the good progostic group compared to the poor prognostic group (511.0 versus 329.0; *p*-value = 0.02). Similarily, median survival is higher in the non-REP versus the REP group (474.5 versus 172.0; *p*-value 0.006). Moreover, a fractal texture feature (T1C_mBm_GLZSM_LargeZoneLowGrayEmphasis) is found significant (*p*-value < 0.05), along a with histogram mean (edema tumor region) feature, for the stratification of prognostic groups as presented in Figure 7. As for incorporating molecular data, the inclusion of IDH mutation status with the radiomics features is significantly associated with survival as shown in Figure 5. Despite this association being reported in the literature [15,40,41], our analysis shows that MGMT promoter methylation status is not associated with survival outcomes (*p*-value = 0.9651), similar to that in the literature [8]. This may be explained by the fact that MGMT promoter methylation status is missing in 14 patients.

Our findings are consistent with earlier research employing sophisticated imaging features, including conventional shape, volumetric, histogram statistics, texture, and fractal-based texture features, as well as radiogenomics, in brain tumor segmentation, classification, survival prediction, and molecular mutation characterization [10,28,42]. Several studies focus on the application of machine or deep learning models for survival predicition and psudeoprogression prediction with radimics features extracted from structuralor advanced MRI [10,43,44,45,46]. The inclusion of fractal-based features with conventional radiomics features in the fractal model increases predictive performance significantly compared to using only conventional radiomics features in the non-fractal model. The feasibility of using fractal-based features is also observed in prognostic grouping using the CGestimator. The percentage of REP cases is higher (84.62%, 11 out of 13) with fractal features-based PI compared to non-fractal feature-based PI (76.92%, 10 out of 13). Futhermore, a multi-fractal-based texture feature extracted from T1C is signifiant in prognostic grouping and REP status stratification.

Limitations of our study include the indeterminate status of molecular markers for some patients. The molecular marker of 1p/19q co-deletion status is diagnostic for oligodendroglioma which is a different type of glioma, so it is not included in our analysis. The indeterminate status of MGMT methylation and IDH mutation may also impact the analysis of these molecular markers. Moreover, the sample size of our study is rather small and may restrict the generalizability of this study. Attempts have been made to address these challenges. Random sampling with 5-fold cross-validation has been performed for feature selection and model evaluation to balance the number of patient cases in each iteration. In addition, only statistically significant features have been included in REP classification and survival modeling. For survival analysis, copula modeling has been utilized to circumvent the assumption of independent censoring.

Future studies with a larger sample size and MRIs from different institutions/scanners are needed for improved generalizability of our model. Including additional sequences (i.e, T2/FLAIR) in addition to T1C may improve REP modeling, both in pre-operative and radiation-planning scans. The ability to predict who will develop rapid progression and where spatially using pre-operative scans can provide valuable insights for guiding surgical and radiation-planning decisions. 

## 5. Conclusions

The most aggressive glioma in adults is GB, and REP is a negative prognostic factor. Our study demonstrates that utilizing both conventional and sophisticated multi-resolution fractal image features from the radiation-planning T1C MRI sequence provides a useful tool for predicting REP in GB patients. Additionally, the copula-based feature selection modeling and survival analysis under dependent censoring indicate the feasibility of fractal and radiomics features for predicting REP in GB patients utilizing radiation-planning MRIs.

## Figures and Tables

**Figure 1 cancers-15-04636-f001:**
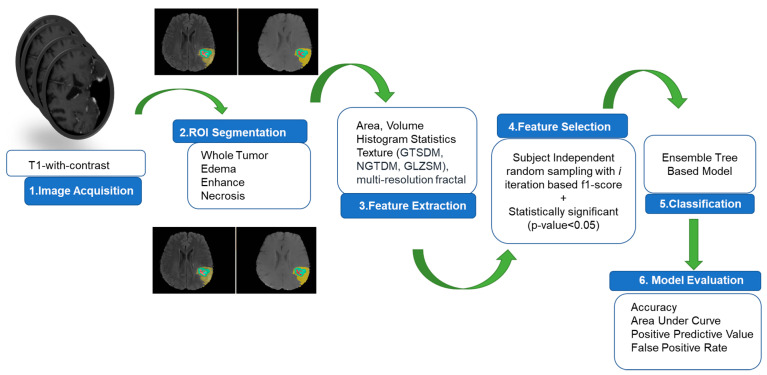
Overall pipeline for REP prediction.

**Figure 2 cancers-15-04636-f002:**
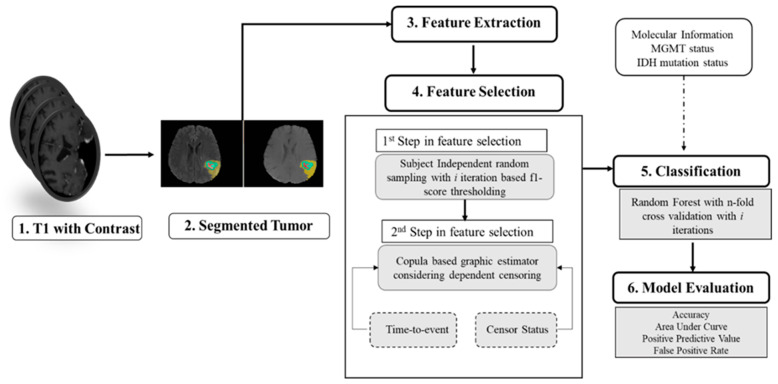
Overall pipeline for prediction of survival.

**Figure 3 cancers-15-04636-f003:**
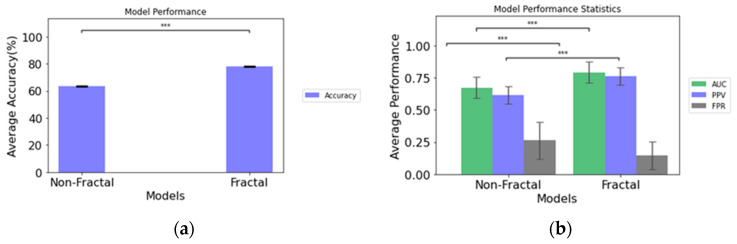
(**a**) Accuracy of the non-fractal and fractal models; (**b**) performance statistics of the non-fractal and fractal models. Error bars represent the standard deviation and asterisks (***) depicts significant differences between the model configurations.

**Figure 4 cancers-15-04636-f004:**
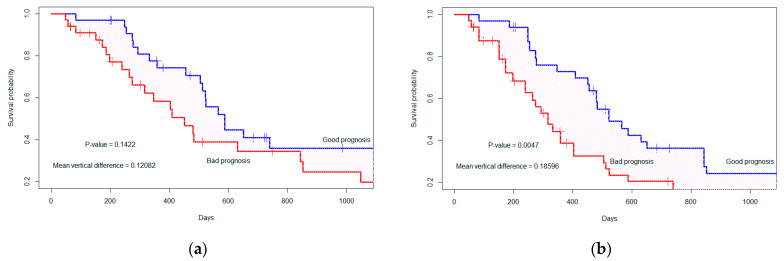
The marginal survival curve is separated by the top two significant features considering (**a**) independent censoring (where α = 0), *p*-value= 0.1422; (**b**) dependent censoring (where α = 18, c-index = 0.519), *p*-value = 0.0047. The ‘+’ indicates the censored cases.

**Figure 5 cancers-15-04636-f005:**
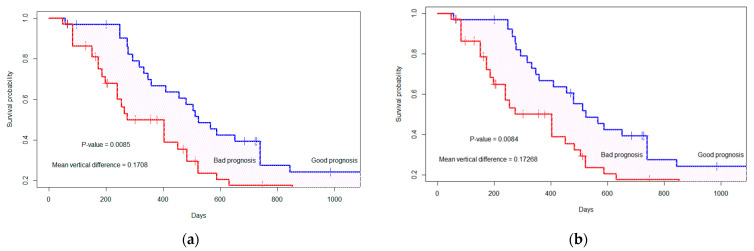
The marginal survival curve for good (or bad) prognosis group is separated by top seven features+ molecular status (IDH mutation). (**a**) Fractal-molecular model; (**b**) non-fractal molecular model. The ‘+’ indicates the censored cases.

**Figure 6 cancers-15-04636-f006:**
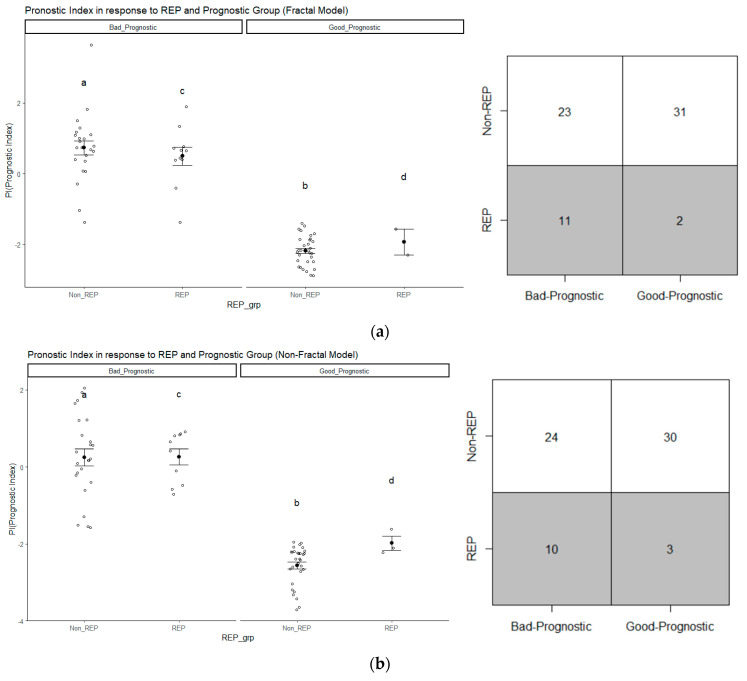
The distribution of REP progression cases in (**a**) the fractal model and (**b**) the non-fractal model based on prognostic group (good or bad); the group labeled “c” corresponds to the patients in the bad prognostic group with REP status, similarly “a” (Non-REP + high PI), “b” (Non-REP + low PI), “d” (REP + low PI).

**Figure 7 cancers-15-04636-f007:**
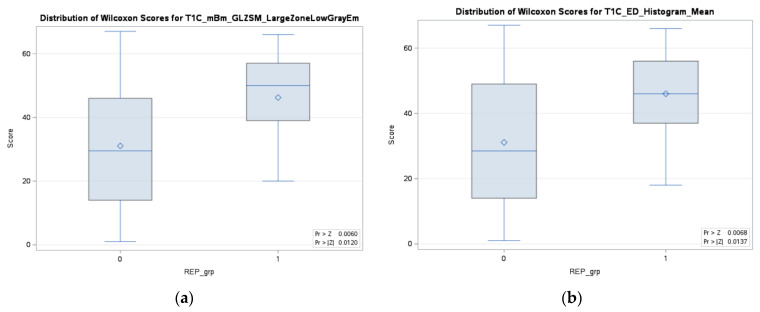
Two significant (*p*-value < 0.05) features for both survival probability under dependent censoring and REP progression status in the fractal model, (**a**) multifractal feature from T1C, and (**b**) histogram mean from edema region of tumor tissue.

**Table 1 cancers-15-04636-t001:** Summary of data between the patient group.

	Total (*n* = 70)	REP (*n* = 13)	Non-REP (*n* = 57)
Survival Days from Surgery			
Present (patient dead/expired)	45	8	37
Lost follow-up (censored) *	22	5	17
Not dead nor censored	3	0	3
MGMT Promoter Status			
Hypermethylated	23	5	18
Unmethylated	33	6	27
Indeterminate	14	2	12
IDH-1 Mutation Status			
Wild type	59	12	47
Mutant	8	1	7
Indeterminate	3	0	3
1p-19q-Codeletion Status			
Codeletion	2	0	2
Negative	18	1	17
Indeterminate	50	12	38

* Some of the censored patients are expired (dead); therefore, all the censored patients are not alive.

**Table 2 cancers-15-04636-t002:** Stratification of patients’ censored and expiration status.

	Expired/Dead (Denoted as 1)	Alive (Denoted as 0)
Non-Censored Patients (*n* = 45)	45	None
Censored/Lost Follow-up Patients (*n* = 22)	9	13

**Table 3 cancers-15-04636-t003:** Comparison between model configurations for 5-fold cross--validation over 1000 iterations with subject independent random sampling for REP classification.

Model Configurations	Area under Curve (AUC)	Accuracy (%)	Positive Predictive Value (PPV)	False Positive Rate (FPR)
Non-Fractal Model	0.673 ± 0.082	63.5 ± 0.069	0.617 ± 0.067	0.262 ± 0.177
Fractal Model	0.793 ± 0.082	78.1 ± 0.071	0.761 ± 0.069	0.145 ± 0.107

**Table 4 cancers-15-04636-t004:** Significant features using copula modeling (dependent censoring). The features are ordered according to *p*-value.

**Fractal Model Features**		
**Features Name**	**Co-Efficient**	***p*-Value**
ET2 ^1^	−1.58	0.0045
T1C_ptpsa_GLZSM_Low_Gray_Level_Zone_Emphasis	1.33	0.0110
L2_Orientation ^2^	0.74	0.0183
edema_FirstAxisLength ^3^	−0.91	0.0194
wt_MajorAxisLength ^4^	−0.93	0.0198
L1_Extent ^2^	0.76	0.0218
L3_Orientation ^2^	0.70	0.0261
T1C_mBm_GLZSM_LargeZoneLowGrayEmphasis	2.27	0.0316
nec_SecondAxis_1 ^5^	−1.06	0.0355
T1C_ED_Histogram_Mean ^6^	1.09	0.0434
**Non-Fractal Model Features**		
**Features Name**	**Co-Efficient**	***p*-Value**
ET2 ^1^	−1.58	0.0045
L2_Orientation ^2^	0.74	0.0183
edema_FirstAxisLength ^3^	−0.91	0.0194
wt_MajorAxisLength ^4^	−0.93	0.0198
L1_Extent ^2^	0.76	0.0218
L3_Orientation ^2^	0.70	0.0261
nec_SecondAxis_1	−1.06	0.0355
T1C_ED_Histogram_Mean ^6^	1.09	0.0434
ED_up_left_y *	0.83	0.0585
T1C_ED_Histogram_Skewness *	−1.18	0.0718

* Features are not significant, only eight features are significant in non-fractal configurations. ^1^ Eccentricity of whole tumor region; ^2^ L1, L2, L3 indicates *x*, *y*, *z* axis of whole tumor region; ^3^ major or first axis length from edema region, ^4^ major or first axis length of whole tumor region, ^5^ second or *y*-axis length of necrotic region, and ^6^ histogram statistics of edema region with a T1C sequence.

**Table 5 cancers-15-04636-t005:** Comparison across model configurations of mean test performance across 5-fold with 1000 iterations for binary prediction of patient survival (patient expired (dead) or not).

Number of Features	Model Configurations	Area under Curve (AUC)	Accuracy (%)	Positive Predicted Value (PPV)	False Positive Rate (FPR)
Top 3 features	Non-Fractal Model	0.730 ± 0.235	74.018 ± 0.045	0.843 ± 0.054	0.351 ± 0.088
	Fractal Model	0.659 ± 0.241	67.67 ± 0.040	0.817 ± 0.058	0.453 ± 0.088
Top 5 features	Non-Fractal Model	0.783 ± 0.199	73.22 ± 0.048	0.847 ± 0.057	0.356 ± 0.098
	Fractal Model	0.658 ± 0.243	70.03 ± 0.048	0.811 ± 0.059	0.420 ± 0.090
Top 7 features	Non-Fractal Model	0.735 ± 0.219	73.05 ± 0.045	0.872 ± 0.054	0.339 ± 0.106
	Fractal Model	0.762 ± 0.214	74.39 ± 0.046	0.881 ± 0.056	0.311 ± 0.109
Top 9 features	Non-Fractal Model	0.725 ± 0.224	71.00 ± 0.049	0.861 ± 0.059	0.378 ± 0.115
	Fractal Model	0.719 ± 0.214	70.37 ± 0.048	0.844 ± 0.061	0.397 ± 0.107

**Table 6 cancers-15-04636-t006:** Comparison across model configurations (top seven features with molecular status (IDH mutation) of mean test performance across 5-fold with 1000 iterations for binary prediction of patient survival (patient expired (dead) or not).

Model Configurations	Area under Curve	Accuracy (%)	Positive Predictive Value (PPV)	False Positive Rate (FPR)
Non-Fractal Molecular	0.757 ± 0.214	72.36 ± 0.046	0.866 ± 0.058	0.354 ± 0.109
Fractal Molecular	0.762 ± 0.214	73.48 ± 0.047	0.883 ± 0.057	0.322 ± 0.114

**Table 7 cancers-15-04636-t007:** Statistical analysis of Survival time (includes censoring time) in prognostic and REP groups for the fractal model.

Group Name	Number of Cases	Mean	Standard Deviation	Standard Error	Median	Range	*p*-Value
Bad Prognostic	34	420.382	335.353	57.513	329.00	48.00–1211.00	0.02
Good Prognostic	33	678.424	494.209	86.031	511.00	57.00–1821.00	
Non-REP	54	604.259	441.961	60.143	474.50	48.00–1821.00	0.006
REP	13	311.615	340.626	94.472	172.00	57.00–1211.00	

## Data Availability

Data may be available subject to appropriate IRB approval.

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
