# Peer review of "Prediction of Rapid Early Progression and Survival Risk with Pre-Radiation MRI in WHO Grade 4 Glioma Patients"

_cancers, 2023, doi:10.3390/cancers15184636_

Round 1

Reviewer 1 Report

Your work is on an interesting topic for possible future developement of therapeutic strategies in this severe disease of GBM, however for the typical clinician reader of the journal the paper is very hard to read and methodology and models difficult to understand. This is why i suggest a major revision of the manuscript. 

Introduce an appendix for detailed description of the models used, algorithms, fomula sections and shorten the manuscript focussing on the clinical relevant parts.

Figures 6 and 8 necessary?

Several typing or spelling errors should be corrected:

line 63: eliminate comma

line 69: has

line 73: lost to follow-up

line 90: predictive

line 94: data were abstracted

line 173: deviation deviations ?

Good quality of english language, only minor corrections and typing errors as mentioned above 

Reviewer 2 Report

This article discusses the feasibility of radiomics and a sophisticated multi-resolutional fractal texture features to predict rapid early progression status in glioblastoma patients from radiation-planning T1C sequence MRI scans. Two models (non-fractal and fractal) are constructed utilizing a random sampling 5-fold cross validation process. The novelty of the work is highlighted by the fact that, based on the data presented, there appears to be a significant difference between the fractal and the non-fractal model in the prediction of rapid early progression status.

In my experience, 10-fold cross validation tends to be more popular than 5-fold validation in studies such as these. Why did the authors chose it?

Line 51: comma formatting error

Line 52: Novo-TTF?

Line 56: Grammar: “There is almost a 50%”

Line 56-57: formatting (spaces and font)            

Line 58: Grammar: “In the evaluation”. In general, I found that little filler words like “the” have often been left out – I’m assuming this is a stylistic choice. 

 Line 63-64: EP is often looked at in comparison to Pseudoprogression, so I feel as though it would have been nice to emphasise this relationship a little more just through wording choices? Something like “The comparison of the first post-radiation MRI and the early post-operative MRI, or more commonly to pre-radiation, is often made in order to evaluate tumour progression as well as radiation induced changes (collectively termed as pseudoprogression)”.

Line 136: How many ROs with what experience? Transfer learning is necessary but not great with this kind of research due to the features being so sensitive so it would have been nice to have this method justified/explained a little better. 

Line 138: Tense: “are”, “include”. Previous paragraph was in past tense whereas this paragraph was in present. So is the next section.

 Line 152: Wording. I think it makes more sense to say something like: “These model configurations include a) a non-fractal model, and b) a fractal model”.

 Line 156: ‘Denotes” used too many times in a row. Not sure too much about bootstrap algorithms so not much help here.

 Lines 164-165: Wording. Justifying why 0.6 was the threshold value was done confusingly

 Line 170: Why only 3 features?

Table 6: Since there are many ways to extract the radiomics features and many different features that can be extracted, it would have been good for the authors to explain how they extracted their features (e.g. using pyradiomcis open source package etc so others can compare their results).  

Line 594: Formatting error: “Ref[8]”

Many grammatical (especially the omission of articles) and spelling errors need to be corrected. I have highlighted some but there are many others. I suggest the authors consult the assistance of a professional editing service for assistance with this task.

Reviewer 3 Report

Rapid early progression in patients with gliomas occurs.  In this paper a random sampling method was developed to predict those cases of more rapid progression.  The results demonstrate the efficacy of pre-radiation image features in the analysis of survival probability and overall prognosis.  The experimental results further show that multi-resolution fractal texture features perform better than conventional radiomics features for prediction of rapid progression and survival outcomes.  This is an interesting paper of limited clinical value in that in general these patients do poorly, and it is unclear how management would be affected by the knowledge that a certain subset of patients will progress more rapidly given that there are limited treatment options.

Round 2

Reviewer 1 Report

After revision the manuscript has been improved - now publication in cancers is recommended